# Antimicrobial Resistance of *Enterobacteriaceae* in Bloodstream Infections in Hospitalized Patients in Southern Poland

**DOI:** 10.3390/jcm11143927

**Published:** 2022-07-06

**Authors:** Marta Kłos, Estera Jachowicz, Monika Pomorska-Wesołowska, Dorota Romaniszyn, Grzegorz Kandzierski, Jadwiga Wójkowska-Mach

**Affiliations:** 1Department of Molecular Biology, Faculty of Science and Health, Catholic University of Lublin, 20-708 Lublin, Poland; 2Department of Microbiology, Faculty of Medicine, Medical College, Jagiellonian University, 31-121 Kraków, Poland; estera.jachowicz@gmail.com (E.J.); varicella_zoster@interia.pl (D.R.); jadwiga.wojkowska-mach@uj.edu.pl (J.W.-M.); 3Department of Microbiology, Analytical and Microbiological Laboratory (KORLAB), 41-700 Ruda Śląska, Poland; monikapw@op.pl; 4Department of Pediatric Orthopedics and Rehabilitation, Medical University of Lublin, 20-093 Lublin, Poland; grzegorz.kandzierski@uszd.lublin.pl

**Keywords:** antimicrobial resistance, bloodstream infections, ESBL-producing *Enterobacteriaceae*

## Abstract

**Aim:** The aim of this study was to highlight antimicrobial resistance among *Enterobacteriaceae* isolated from bloodstream infections in hospitals in southern Poland. **Materials and Methods:** The present study includes laboratory-confirmed secondary bloodstream infections (LC-BSIs), in the years 2015–2018, in hospitalized adult patients (≥18). Episodes of BSIs were defined according to the strictly described guidelines. Antimicrobial susceptibility testing was performed with the automated system and the disc diffusion method. Extended-spectrum β-lactamases (ESBLs)-producing *Enterobacteriaceae* were detected using the double-disc synergy test. **Results:** Between 2015 and 2018, 356 episodes of secondary BSIs in 997 patients aged 21–96 years were documented in a prospective study, including 134 (37.6%) ESBL-producing *Enterobacteriaceae*. *Escherichia coli* was the predominant pathogen in internal medicine (37.6%) and surgery units (46.8%); in intensive care units (ICUs), *Klebsiella pneumoniae* was isolated more frequently (33.3%). *Enterobacteriaceae* were highly resistant to most antimicrobial agents. *K. pneumoniae* isolates had a higher level of resistance than *E. coli*, regardless of the unit. **Conclusions:** The increase in AMR and the widespread distribution of ESBL-producing *Enterobacteriaceae* in Polish hospitals can be related to the lack of or inappropriate antibiotic treatment.

## 1. Introduction

As reported by the Centers for Disease Control and Prevention (CDC), hospital-acquired infections (HAIs) are defined as systemic conditions resulting from an adverse reaction to the presence of an infectious agent(s) or its toxin(s) [1]. Healthcare-associated infections (HCAIs) refer, for example, to hospital care, long-term care, home care, and ambulatory care [2]. These terms, ‘healthcare-associated infection (HCAI),’ and ‘hospital-acquired infection (HAI),’ are used interchangeably. According to the European Centre for Disease Prevention and Control (ECDC), each year in the European Union (EU), approximately 4 million patients acquire a healthcare-associated infection [3].

HAIs, including BSIs, entail intensive antimicrobial treatment that can cause an adverse reaction, including the acquisition of bacterial resistance to antimicrobials. Antimicrobial resistance (AMR) and bloodstream infections (BSIs) have been an increasing challenge in healthcare due to the deficiency of effective antimicrobials. BSI is termed a potentially life-threatening condition characterized by high morbidity and mortality, with a mortality rate that reaches 40% in ICUs [4,5]. In Poland, the mortality was 20% among patients with BSIs in ICUs [6]. Furthermore, BSIs extend hospitalization time and increase healthcare costs; in Poland, the average hospitalization time in the ICU was up to 34 days [7].

The multidrug-resistance (MDR) phenotype is associated with the production of TEM, SHV, and CTX-M enzymes in *Enterobacteriaceae*; it has been associated with nosocomial infections [8,9]. Multidrug-resistant strains, inappropriate prophylaxis, and insufficient or delayed antibiotic therapy result in the failure of empiric antimicrobial therapy [10]. The effectiveness of antibiotic prophylaxis should be considered. Although antibiotic prophylaxis is known to be essential in surgical procedures, some mechanisms of AMR are still unknown, and increasing AMR in *Enterobacteriaceae* makes treatment difficult. To reduce the risk of spreading drug resistance, intensive work is being carried out on the development of standards for the use of antibiotics. The Antibiotic Stewardship Program (ASP) consists of promoting the principles of rational use of antibiotics: the selection of the optimal antibiotic, the appropriate dose and route of administration, and the duration of treatment.

The aim of this study was to highlight the high levels of antimicrobial resistance among *Enterobacteriaceae* isolated from bloodstream infections in hospitals in southern Poland, with special emphasis on extended-spectrum β-lactamases (ESBLs)-producing *Enterobacteriaceae*.

## 2. Materials and Methods

The present study included laboratory-confirmed secondary bloodstream infections (LC-BSIs) that occurred between 1 January 2015 and 31 December 2018 in hospitalized adult patients (≥18) in southern Poland (13 hospitals). An episode of BSI was defined according to the ECDC guidelines [11,12]. Furthermore, according to the guidelines and the study inclusion criteria of secondary BSIs: (1) a blood culture positive for ESBL-producing *Enterobacteriaceae*; (2) the pathogen identified from the blood sample matched the pathogen identified from the site of infection; (3) blood samples were collected from separate venous punctures, avoiding collection through the catheter, in order to reduce the risk of a diagnostic error [1,13]. The patient’s health status was evaluated by the attending physician, based on clinical data, directly in the hospital. If more than one organism was isolated from one patient or blood culture, only the first isolate from one patient was included in the analysis of the same infection case.

Blood cultures were performed in BACTEC Plus Aerobic/F Culture Vials and BACTEC Plus Anaerobic/F Culture Vials (Becton Dickinson Diagnostic Instrument Systems, Sparks, MD, USA), interchangeable with BACTEC Lytic/10 Anaerobic Culture Vials (Becton Dickinson Diagnostic Instrument Systems, Sparks, MD, USA). The isolates were identified by matrix assisted laser desorption ionization time of flight mass spectrometry (MALDI-TOF MS Biotyper; Bruker Corporation, MA, USA) according to standard methods [14]. The presence of the *bla* genes for ESBLs (TEM, SHV, CTX-M) was determined by multiplex PCR using specific primers [15]. Antimicrobial susceptibility testing was performed by the disc diffusion method (Oxoid Thermo Scientific, Basingstoke, UK) and with the automated BD Phoenix^TM^ 100 system (Becton Dickinson Company, Sparks, MD, USA) [16]. The NMIC/ID-204 Panels (Becton Dickinson Company, Sparks, MD, USA) were used to determine antimicrobial susceptibility with the BD Phoenix^TM^ 100 automated system according to the manufacturer’s instructions [16]. The results of antibacterial susceptibility testing were interpreted according to breakpoints established by the European Committee on Antimicrobial Susceptibility Testing (EUCAST) in 2022 [17]. In this study, antimicrobial resistance was determined into six categories: aminoglycosides, carbapenems, cephalosporins, fluoroquinolones, glycopeptides and lipoglycopeptides, and macrolides. Microbes intermediately susceptible to antibiotics were classified as resistant, as only susceptible microbes can be regarded as being able to be managed by means of the respective antibiotic agent [18]. ESBL-producing strains were detected using the double-disc synergy test, according to the procedure of Kaur et al., with cefotaxime, ceftazidime, cefepime, and aztreonam placed around the disc with amoxicillin and clavulanic acid [19].

### Statistical Analyses

The relationships between the studied groups were tested by Pearson’s chi-square test: if Cochran’s condition was met, but if the condition was not met, Fisher’s exact test was used. The Yates amendment was applied when the observed numbers were <10. It was assumed to be statistically significant if *p* < 0.05, and the test probability was statistically significant if *p* < 0.05, and it was assumed that the test probability was highly statistically significant if *p* < 0.01. The analyses were performed with PQStat Software version 1.8.0.444 (Poland).

Ethic code: The study was approved by the Jagiellonian University Bioethics Committee no. 1072.6120.64.2019, date: 28 March 2019.

## 3. Results

In total, during the study period 2015–2018, 356 *Enterobacterales* were isolated. As presented in Table 1, *Escherichia coli* was most often isolated strain in internal medicine units (140; 37.6%) and surgery units (74; 46.8%). In intensive care units (ICUs), the predominant species was *Klebsiella pneumoniae* (9; 33.3%). A highly significant association (*p* < 0.01) in the prevalence of *K. pneumoniae* in different units was found.

Table 2 shows that half of *Klebsiella* spp. were extended-spectrum β-lactamases (ESBLs)-producing, both in the internal medicine and surgical units. The ESBL-positive phenotype was detected in most *Enterobacterales* (61.5%) from the “Others” group, isolated in internal medicine units. The genus *Klebsiella* included the species *Klebsiella oxytoca* and *Klebsiella mobilis*. In the “Others” group, the species *Enterobacter cloacae*, *Proteus mirabilis,* and *Citrobacter freundii* were identified.

As summarized in Table 3, regardless of unit or antimicrobial agents, *K. pneumoniae* isolates were more resistant than *E. coli*. In each of the units studied, high resistance of the strains to ampicillin and ampicillin-sulbactam was observed (up to 100%). Among the high resistance of *K. pneumoniae* isolates to tested antimicrobials, a high resistance rate was observed for ciprofloxacin (up to 74.1%), and trimethoprim-sulfamethoxazole (up to 59.2%). The higher susceptibility of *Enterobacteriaceae* isolates was observed using β-lactamase inhibitors, especially tazobactam and sulbactam (depending on the unit and strains, in the range of 6.4–55.6%). Regardless of the unit, resistance to amikacin was the lowest (up to 40.7%), and resistance to tobramycin was the highest (up to 55.6%). No statistically significant association (*p* < 0.05) was found with *Enterobacteriaceae* antimicrobial resistance in dissimilar units.

Overall, 134 (37.6%) episodes of HCA-BSIs caused by ESBL-producing *Enterobacteriaceae* were documented during the research: of these, the predominant were CTX-M-producing strains, regardless of the unit. ESBL-producing *E. coli* and ESBL-producing *K. pneumoniae* were the most frequently detected regardless of units (in internal medicine units 31.3% and 17.2%, respectively), due to this, these strains were analysed for antimicrobial resistance. ESBL-producing *Enterobacteriaceae* were more resistant than non-ESBL *Enterobacteriaceae* to most antimicrobial categories, including penicillins (up to 100%), cephalosporins (up to 75.0%), fluoroquinolones (up to 83.3%), and trimethoprim-sulfamethoxazole (up to 75.0%). The highest antimicrobial resistance was reported in ICUs, among ESBL-producing *E. coli*, reaching 100%. No statistically significant association (*p* < 0.05) was found between ESBL-producing *Enterobacteriaceae* antimicrobial resistance to different antimicrobials in dissimilar units.

## 4. Discussion

Overall, during the 2015–2018 study period, 557 bacterial strains originating from secondary bloodstream infections were analysed, including 161 (29.0%) Gram-positive cocci, 356 (63.8%) *Enterobacterales,* and 40 (7.2%) Gram-negative bacilli other than *Enterobacterales*. The highest percentage of isolated *Enterobacteriaceae* was *Escherichia coli* (222; 39.9%), regardless of unit. The results obtained indicate significant participation of *Enterobacterales*, which represented almost 2/3 of the aetiology of secondary BSIs, which is a very disturbing result and should become the basis for a comprehensive discussion of the condition supervision of infections in Polish hospitals. The results of the SENTRY Antimicrobial Surveillance Program show that after 2005, *E. coli* was the predominant species isolated from bloodstream infections (BSIs) [20]. In addition to *E. coli*, *Staphylococcus aureus* was the second predominant strain, depending on the geographic region, the type of infection, and age [20]. The SENTRY Antimicrobial Surveillance Program monitors the predominant bacterial pathogens and the antimicrobial resistance of isolated organisms from patients with various infection types, including BSI [20]. The SENTRY Antimicrobial Surveillance Program reports trends in organism distribution and antimicrobial resistance (AMR) among BSI isolates submitted to the SENTRY Program [20]. In the years 2013–2016, a particular increase in the percentage of *E. coli* and *Klebsiella pneumoniae* has been observed in BSI episodes in Europe and Asia [21]. Independent research report that in intensive care units (ICUs), *Enterobacteriaceae* were one of the most often isolated strains [22,23]. According to the European Centre for Disease Prevention and Control (ECDC) data for 2014 and 2017, one of the most common isolated strains of ICUs, was *Klebsiella* spp., which shows a constant trend for the most frequently isolated pathogens of BSIs [24,25].

The main findings obtained in this study are confirmed by the ECDC report published in 2019 on antimicrobial resistance (AMR) in hospital BSIs: in Poland, the percentage of *E. coli* isolates resistant to aminopenicillins is one of the highest in Europe and amounts to 61.6%, as in other countries of southern Europe or western and northern Europe [26]. Ghadiri et al. also confirm the high resistance of *E. coli* isolates to ampicillin (63.2%) and ciprofloxacin (47.4%) [27]. The EUROBACT International Cohort Study, conducted in 2012, covering 162 ICUs in 24 countries, showed a high percentage (47.8%) of multidrug-resistant microorganisms, including 7.4% *E. coli* and 11.9% *K. pneumoniae* [28]. Resistance to aminoglycosides remains a contentious issue, depending on the place of acquisition, *Enterobacteriaceae* isolates, and antimicrobial agents: as the results show, in internal medicine and ICUs, resistance to amikacin did not exceed 25.0%, while in surgical units it reached even 40.7% (referring to *K. pneumoniae*). Compared to other aminoglycoside antibiotics, regardless of unit, resistance to tobramycin was highest and reached 55.6% (in ICUs). In surgical units, a different percentage of resistant isolates related to gentamicin have also been reported, although not exceeding 50.0%. According to the ECDC data, high resistance to cephalosporins was observed, particularly among *E. coli* isolates (17%); however, the percentage of cephalosporins-non-susceptible *E. coli* was significantly lower than the results obtained in this study (up to 50.0% *E. coli*, regardless of the unit, referring to cephalosporins without β-lactamase inhibitors) [29]. In relation to *K. pneumoniae* isolates, Poland has one of the highest antimicrobial resistance results among the data from 33 countries surveyed, referring to fluoroquinolones (61.3%, in this study *K. pneumoniae* up to 74.1%), aminoglycosides (47.5%, in this study *K. pneumoniae* up to 55.6%), cephalosporins (58.3%, in this study *K. pneumoniae* up to 66.7%) [26]. The increase in antimicrobial resistance (AMR) of *Enterobacteriaceae* and the widespread distribution of ESBL-producing *Enterobacteriaceae* in Polish hospitals can be related to the lack of or inappropriate antibiotic treatment. Hence, it has been reducing the number of antimicrobial agents to which the pathogen is susceptible and the available treatment options.

A total of 134 extended-spectrum β-lactamases (ESBLs)-producing *Enterobacteriaceae* (37.6%) isolates were enrolled; of these *E. coli* isolates, 48.5%, and *K. pneumoniae* isolates 35.8%. Kallel et al. indicate a relatively widespread number of ESBL-producing isolates and a high percentage of ESBL-producing *Enterobacteriaceae* strains (27.6%) [23]. The occurrence of an ESBL-positive phenotype is associated with nosocomial infections [8,9] but is also related to the multidrug-resistance (MDR) of *Enterobacteriaceae* [8]. In the years 2012–2014, 15,588 *Enterobacteriaceae* isolates were tested in hospitals in the USA and 13.7% of them were ESBL-positive, demonstrating the difference between the results obtained in this study [30].

## 5. Conclusions

Among *Enterobacteriaceae* isolates, *Klebsiella pneumoniae* was more resistant than *Escherichia coli*. *Enterobacteriaceae* were highly resistant to most antimicrobial categories, including β-lactams, fluoroquinolones, and trimethoprim-sulfamethoxazole. Extended-spectrum β-lactamases (ESBLs)-producing *Enterobacteriaceae* were detected more frequently in internal medicine units (*E. coli* 64.6% and *K. pneumoniae* 47.9%). The high resistances of isolates suggest a problem with choosing the drug for empirical treatment. According to our results, only three antibiotics are likely to be effective in empirical treatment: piperacillin-tazobactam, cefoperazone-sulbactam, and amikacin. The high percentage of bloodstream infections (BSIs) and high resistance of the isolated strains suggest inadequate standards for infection control, increased multidrug-resistant microbes, and the need for the implementation and improvement of the Antibiotic Stewardship Program. Periodic training on the above-mentioned issues for healthcare workers may also be irreplaceable.

## Figures and Tables

**Table 1 jcm-11-03927-t001:** Aetiology of BSI episodes.

		Type of Hospital Unit			
		Internal Medicine Units	Surgery Units	ICUs	Overall
**non-*Enterobacterales n* = 161**	Gram-positive cocci				**161 (29.0%)**
	*Staphylococcus aureus*	100 (26.9%)	12 (7.6%)	5 (18.5%)	117 (21.0%)
	*Enterococcus faecalis*	17 (4.6%)	3 (1.9%)	2 (7.4%)	22 (3.9%)
	*Enterococcus faecium*	3 (0.8%)	1 (0.6%)	0	4 (0.7%)
	*Streptococcus agalactiae*	2 (0.5%)	0	0	2 (0.3%)
	Group G Streptococci	1 (0.3%)	2 (1.3%)	0	3 (0.5%)
	*Streptococcus constellatus*	1 (0.3%)	0	0	1 (0.2%)
	*Streptococcus pyogenes*	1 (0.3%)	0	1 (3.7%)	2 (0.3%)
	*Coagulase-negative staphylococci*	8 (2.1%)	2 (1.3%)	0	10 (1.8%)
** *Enterobacterales* ** ***n* = 356**	Gram-negative bacilli				**356 (63.8%)**
	*Escherichia coli*	140 (37.6%)	74 (46.8%)	8 (29.6%)	222 (39.9%)
	*Citrobacter koseri*	1 (0.3%)	0	0	1 (0.2%)
	*Citrobacter freundii*	0	1 (0.6%)	0	1 (0.2%)
	*Citrobacter braakii*	1 (0.3%)	0	0	1 (0.2%)
	*Enterobacter cloacae*	10 (2.7%)	3 (1.9%)	0	13 (2.3%)
	*Klebsiella pneumoniae*	44 (11.8%)	27 (17.1%)	9 (33.3%)	80 (14.4%)
	*Klebsiella oxytoca*	6 (1.6%)	5 (3.2%)	0	11 (2.0%)
	*Klebsiella mobilis*	1 (0.3%)	1 (0.6%)	0	2 (0.3%)
	*Salmonella enteritidis*	1 (0.3%)	0	0	1 (0.2%)
	*Serratia marcescens*	1 (0.3%)	1 (0.6%)	0	2 (0.3%)
	*Morganella morganii*	3 (0.8%)	0	0	3 (0.5%)
	*Proteus mirabilis*	9 (2.4%)	10 (6.3%)	0	19 (3.4%)
**Other *n* = 40**	Gram-negative bacilli; other				**40 (7.2%)**
	*Acinetobacter baumanii*	9 (2.4%)	0	1 (3.7%)	10 (1.8%)
	*Acinetobacter Iwoffii*	1 (0.3%)	0	0	1 (0.2%)
	*Pseudomonas aeruginosa*	10 (2.7%)	16 (10.1%)	1 (3.7%)	27 (4.8%)
	Other	2 (0.5%)	0	0	2 (0.3%)
	**Overall**	**372 (100%)**	**158 (100%)**	**27 (100%)**	**557 (100%)**

**Table 2 jcm-11-03927-t002:** Prevalence of ESBL-producing *Enterobacterales* in BSI episodes according to the place of acquisition.

	Type of Hospital Unit			
ESBL-Producing *Enterobacterales n* = 134	Internal Medicine Units	Surgery Units	ICUs	Overall
** *Escherichia coli* ** ***n* = 65**	42 (64.6%)	19 (29.2%)	4 (6.2%)	65 (48.5%)
** *Klebsiella* ** ***pneumoniae n* = 48**	23 (47.9%)	18 (37.5%)	7 (14.6%)	48 (35.8%)
** *Klebsiella* ** **spp. *n* = 8**	4 (50.0%)	4 (50.0%)	0	8 (6.0%)
**Others *n* = 13**	8 (61.5%)	5 (38.5%)	0	13 (9.7%)
**Overall**	77 (57.5%)	46 (34.3%)	11 (8.2%)	**134 (100%)**

Legend. ‘Others’ included the species Enterobacter cloacae, Proteus mirabilis, and Citrobacter freundii.

**Table 3 jcm-11-03927-t003:** Antimicrobial resistance of *Enterobacteriaceae* isolated in hospital units.

		Internal Medicine Units *n* = 184		Surgery Units *n* = 101		ICUs *n* = 17	
Antimicrobial Category	Antimicrobial Agent	*E. coli n* = 140 (76.1%)	*K. pneumoniae**n* = 44 (23.9%)	*E. coli n* = 74 (73.3%)	*K. pneumoniae n* = 27 (26.7%)	*E. coli n* = 8 (47.1%)	*K. pneumoniae n* = 9 (52.9%)
**Penicillins**	Ampicillin	133 (95.0%)	43 (97.7%)	73 (98.6%)	27 (100%)	8 (100%)	9 (100%)
	Ampicillin-sulbactam	132 (94.3%)	40 (90.9%)	73 (97.2%)	27 (100%)	8 (100%)	9 (100%)
	Amoxicillin-clavulanic acid	72 (52.1%)	24 (54.5%)	35 (45.9%)	18 (66.7%)	5 (62.5%)	6 (66.7%)
	Piperacillin-tazobactam	24 (17.1%)	15 (34.1%)	10 (13.5%)	9 (33.3%)	1 (12.5%)	5 (55.6%)
**Cephalosporins**	Cefuroxime	56 (40.0%)	20 (45.4%)	29 (39.1%)	13 (48.1%)	4 (50.0%)	6 (66.7%)
	Ceftazidime	52 (37.1%)	19 (43.2%)	29 (39.1%)	12 (44.4%)	4 (50.0%)	6 (66.7%)
	Cefotaxime	53 (37.8%)	19 (43.2%)	28 (37.8%)	11 (40.7%)	4 (50.0%)	6 (66.7%)
	Cefepime	52 (37.1%)	16 (36.4%)	23 (29.7%)	14 (51.8%)	3 (37.5%)	4 (44.4%)
	Cefoperazone-sulbactam	9 (6.4%)	6 (13.6%)	7 (8.1%)	5 (18.5%)	2 (25.0%)	1 (11.1%)
**Fluoroquinolones**	Ciprofloxacin	80 (57.1%)	25 (56.8%)	36 (48.6%)	20 (74.1%)	6 (75.0%)	6 (66.7%)
**Aminoglycosides**	Gentamicin	45 (32.1%)	15 (34.1%)	19 (25.6%)	6 (22.2%)	4 (50.0%)	4 (44.4%)
	Amikacin	19 (13.6%)	11 (25.0%)	11 (14.8%)	11 (40.7%)	2 (25.0%)	1 (11.1%)
	Tobramycin	56 (40.0%)	20 (45.4%)	24 (32.4%)	15 (55.5%)	4 (50.0%)	5 (55.6%)
**Others**	Trimethoprim-sulfamethoxazole	66 (47.1%)	22 (50.0%)	37 (50.0%)	16 (59.2%)	4 (50.0%)	4 (55.6%)
**ESBL-producing *Enterobacteriaceae n* = 134 (37.6%)**		**42 (31.3%)**	**23 (17.2%)**	**19 (14.2%)**	**18 (13.4%)**	**4 (3.0%)**	**7 (5.2%)**

## Data Availability

Data supporting reported results can be found in the Department of Microbiology of Jagiellonian University, including archived datasets analysed or generated during the study.

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
