# Peer review of "Antimicrobial Resistance of Enterobacteriaceae in Bloodstream Infections in Hospitalized Patients in Southern Poland"

_jcm, 2022, doi:10.3390/jcm11143927_

Round 1
Reviewer 1 Report
Overall, this is a well-written, precise, and clear manuscript. The introduction is relevant and theoretical in nature. The prior research findings are described in sufficient detail for readers to understand the current study rationale. The writers provide a systematic contribution to the scientific literature in this field of research. Overall, this is a good manuscript. It requires minor adjustments. Specific remarks are provided below.
A: Introduction:
Second-page 4th line: please mention the name MDR microorganisms that produce enzymes.
B: Materials and methods:
Methods need some clearance and recheck the number of samples.
C: Result:
In the 3rd paragraph name the strain which produces CTX-M.
Please carefully check the all result.
D: Discussion:
Give 2-3 lines about SENTRY Antimicrobial Surveillance Program.
The overall manuscript was written very well.
Author Response
Lublin, 28.06.2022
Concerns the manuscript Antimicrobial resistance of Enterobacteriaceae in bloodstream infections in hospitalized patients in southern Poland
To whom it may concern,
We would like to thank the Reviewer for the time they devoted to the improved manuscript and preparation of comments - all of them were reconsidered.
Below is the full answer to the doubts of the reviewer.
Yours faithfully,
Marta KÅ‚os, Estera Jachowicz, Monika Pomorska-WesoÅ‚owska, Dorota Romaniszyn, Grzegorz Kandzierski, Jadwiga Wójkowska-Mach
Does the introduction provide sufficient background and include all relevant references?
A: Introduction:
Second-page 4th line: please mention the name MDR microorganisms that produce enzymes.
Reply: The introduction was corrected. In the Introduction section, we emphasize the problem of nosocomial infections, including bloodstream infections, increasing mortality, and morbidity. Then we discuss the issue of increasing multidrug-resistance (MDR) of Enterobacteriaceae caused by various factors: inappropriate prophylaxis, lack of effective antimicrobials, failure of empiric therapy. In the Introduction, we note that the multidrug-resistance phenotype (MDR) occurs in Enterobacteriaceae, on which our article is based.
Are the methods adequately described?
B: Materials and methods:
Methods need some clearance and recheck the number of samples.
Reply: We rechecked the number of samples. The difference may be due to the aetiology of bloodstream infections - the result of isolated bacterial strains is given, then divided into individual groups in the table.
Are the results clearly presented?
C: Result:
In the 3rd paragraph name the strain which produces CTX-M.
Please carefully check the all result.
Reply: Thank you for your note. The results have been carefully checked and corrected. The numbers of strains were counted. In the 4th paragraph, we mention that ESBL-producing E. coli and K. pneumoniae are the most frequently isolated strains that produce CTX-M.
D: Discussion:
Give 2-3 lines about SENTRY Antimicrobial Surveillance Program.
Reply: A short description of the SENTRY program was added to the discussion.

Reviewer 2 Report
The manuscript presented to me for review by Marta KÅ‚os et al. entitled "Antimicrobial resistance of Enterobacteriaceae in bloodstream infections in hospitalized patients in southern Poland" presents the crucial topic of drug resistance of microorganisms isolated in hospitals.
This study sheds light on the growing problem of antibiotic resistance and suggests the need for changes in the medical system and raising awareness and qualifications of staff.
Compared to the previous version, this manuscript has significantly improved in terms of the language used and the transparency of the presentation of the results.
Despite numerous corrections and comments from previous reviewers, the manuscript continues to present the results in a very simplified form based on a limited number of results.
I understand that due to the limitations and retrospective nature, it is impossible to enrich the work with additional measurements.
Author Response
Lublin, 28.06.2022
Concerns the manuscript Antimicrobial resistance of Enterobacteriaceae in bloodstream infections in hospitalized patients in southern Poland
To whom it may concern,
We would like to thank the Reviewer for the time they devoted to the improved manuscript and preparation of comments - all of them were reconsidered.
Below is the full answer to the doubts of the reviewer.
Yours faithfully,
Marta KÅ‚os, Estera Jachowicz, Monika Pomorska-WesoÅ‚owska, Dorota Romaniszyn, Grzegorz Kandzierski, Jadwiga Wójkowska-Mach
The manuscript presented to me for review by Marta KÅ‚os et al. entitled "Antimicrobial resistance of Enterobacteriaceae in bloodstream infections in hospitalized patients in southern Poland" presents the crucial topic of drug resistance of microorganisms isolated in hospitals.
This study sheds light on the growing problem of antibiotic resistance and suggests the need for changes in the medical system and raising awareness and qualifications of staff.
Compared to the previous version, this manuscript has significantly improved in terms of the language used and the transparency of the presentation of the results.
Despite numerous corrections and comments from previous reviewers, the manuscript continues to present the results in a very simplified form based on a limited number of results.
I understand that due to the limitations and retrospective nature, it is impossible to enrich the work with additional measurements.
Reply: Thank you for your review and any comments and suggestions. Any additional corrections were made using "Track changes". Our aim was to show the problem of increasing antimicrobial resistance (AMR), and hence the lack of effective therapeutic measures, prophylaxis, and the need for changes in the health system. Therefore, we wanted to show the results of antimicrobial resistance in a simple and clear way. We hope that despite the retrospective nature of the work and the limited number of results, it achieves the intended effect: it shows an extremely high level of antimicrobial resistance of Enterobacteriaceae isolated from bloodstream infections among Polish patients.

Reviewer 3 Report
Antimicrobial resistance of Enterobacteriaceae in bloodstream infections in hospitalized patients in southern Poland
In this study, the authors have analyzed the secondary blood stream infections in hospitalized patients in southern Poland. They have shown that the ESBL producing Enterobacteriaceae are the major pathogens. These E.coli and K.pneumonia were found to be highly resistant to most antimicrobial agents. The results of the study will be useful for scientists working in the area of health care associated infections and antimicrobial resistance. The article is suitable for publication. Corrections suggested below may be carried out before final submission.
Corrections/modifications required
Section |
Line No. |
Corrections |
Abstract |
12 |
K.pneumonia isolates were……..the unit. To be corrected as K.pneumonia isolates had……..the unit. |
Introducion |
14 |
In Poland, ………..ICUs. –Not clear. Please rephrase the sentence. |
Results |
4 |
There were found ……….. dissimilar units. - Not clear. Please rephrase the sentence. |
Note: Corrected/added words are underlined.
Author Response
Lublin, 28.06.2022
Concerns the manuscript Antimicrobial resistance of Enterobacteriaceae in bloodstream infections in hospitalized patients in southern Poland
To whom it may concern,
We would like to thank the Reviewer for the time they devoted to the improved manuscript and preparation of comments - all of them were reconsidered.
Below is the full answer to the doubts of the reviewer.
Yours faithfully,
Marta KÅ‚os, Estera Jachowicz, Monika Pomorska-WesoÅ‚owska, Dorota Romaniszyn, Grzegorz Kandzierski, Jadwiga Wójkowska-Mach
Antimicrobial resistance of Enterobacteriaceae in bloodstream infections in hospitalized patients in southern Poland
In this study, the authors have analyzed the secondary blood stream infections in hospitalized patients in southern Poland. They have shown that the ESBL producing Enterobacteriaceae are the major pathogens. These E.coli and K.pneumonia were found to be highly resistant to most antimicrobial agents. The results of the study will be useful for scientists working in the area of health care associated infections and antimicrobial resistance. The article is suitable for publication. Corrections suggested below may be carried out before final submission.
Corrections/modifications required
Section |
Line No. |
Corrections |
Abstract |
12 |
K.pneumonia isolates were……..the unit. To be corrected as K.pneumonia isolates had……..the unit. |
Introduction |
14 |
In Poland, ………..ICUs. –Not clear. Please rephrase the sentence. |
Results |
4 |
There were found ……….. dissimilar units. - Not clear. Please rephrase the sentence. |
Note: Corrected/added words are underlined.
Reply: Thank you for your review, any comments, and corrections. Suggested corrections and modifications were made/added. The Abstract was corrected as well as rephrased the sentences in the Introduction and the Results. All changes are marked with "Track changes".

This manuscript is a resubmission of an earlier submission. The following is a list of the peer review reports and author responses from that submission.
Round 1
Reviewer 1 Report
The manuscript presented to me for review by Marta KÅ‚os et al. entitled "Antimicrobial resistance of Enterobacteriaceae in bloodstream infections in hospitalized patients in southern Poland" presents the crucial topic of drug resistance of microorganisms isolated in hospitals.
The authors analyzed in detail the data collected over several years, analyzing the antibiotic resistance of microorganisms from the Enterobacteriaceae family.
This study sheds light on the growing problem of antibiotic resistance as well as suggests the need for changes in the medical system and raising awareness and qualifications of staff.
In my opinion, despite its relative simplicity, the manuscript meets the publication criteria by raising an extremely important issue. Because the study was well thought out, and the results were analyzed with due care, based on the applicable diagnostic standards and appropriate statistical tests.
Author Response
Lublin, 10.06.2022
Concerns the manuscript Antimicrobial resistance of Enterobacteriaceae in bloodstream infections in hospitalized patients in southern Poland
To whom it may concern,
We would like to thank the Reviewer for the time they devoted to the improved manuscript and preparation of comments - all of them were reconsidered.
Below is the full answer to the doubts of the reviewer.
Yours faithfully,
Marta KÅ‚os, Estera Jachowicz, Monika Pomorska-WesoÅ‚owska, Dorota Romaniszyn, Grzegorz Kandzierski, Jadwiga Wójkowska-Mach
„English language and style are fine/minor spell check required.”
Reply: The English language and style were checked.
”Does the introduction provide sufficient background and include all relevant references?”
Reply: The introduction was corrected. In the Introduction section, we emphasize the problem of nosocomial infections, including bloodstream infections, increasing mortality and morbidity. Then we discuss the issue of the increasing multidrug resistance of Enterobacterales caused by various factors: inappropriate prophylaxis, lack of effective antimicrobials, failure of empiric therapy.

Reviewer 2 Report
The subject is important but are too many data, confusingly presented. ESBL-producing Enterobacterales and Gram-positive bacteria. It is necessary to separate the data concerning different bacteria of the Enterobacteria family and to present the resistance genes with the genetic mechanisms of the resistance to Carbapenems, to other beta-lactams drugs or to other AB families.
Author Response
Lublin, 10.06.2022
Concerns the manuscript Antimicrobial resistance of Enterobacteriaceae in bloodstream infections in hospitalized patients in southern Poland
To whom it may concern,
We would like to thank the Reviewer for the time they devoted to the improved manuscript and preparation of comments - all of them were reconsidered.
Below is the full answer to the doubts of the reviewer.
Yours faithfully,
Marta KÅ‚os, Estera Jachowicz, Monika Pomorska-WesoÅ‚owska, Dorota Romaniszyn, Grzegorz Kandzierski, Jadwiga Wójkowska-Mach
„English language and style are fine/minor spell check required.”
Reply: The English language and style were checked.
”Does the introduction provide sufficient background and include all relevant references?”
Reply: The introduction was corrected. In the Introduction section, we emphasize the problem of nosocomial infections, including bloodstream infections, increasing mortality and morbidity. Then we discuss the issue of increasing multidrug resistance of Enterobacterales caused by various factors: inappropriate prophylaxis, lack of effective antimicrobials, failure of empiric therapy.
“Are all the cited references relevant to the research?”
Reply: The references cited were checked.
“Are the methods adequately described?”
Reply: The methods were described in detail, including guidelines for bloodstream infection, blood sampling, microbial identification, drug resistance analysis, statistical analysis. Errors were fixed. Ethic code was added. The latest EUCAST guidelines (2022) were used, which were compared with data from 2015-2018, when the study was conducted due to the time of writing.
“Are the results clearly presented?”
Reply: the Results section was improved. The results, after being improved, describe only Enterobacterales antimicrobial resistance with an emphasis on ESBL strains. Only Table 1 presents an overview of the etiology of bloodstream infections among the group of patients studied. Only enzymes from the SHV, CTX-M and TEM groups are mentioned among the ESBL mechanisms. Tables 2 and 3, with the prevalence of ESBL strains, are intended to show their percentage share in individual units.
“Are the conclusions supported by the results?”
Reply: The conclusions were improved.
Reviewer 3 Report
The aim of this study was to highlight antimicrobial resistance among Enterobacteriaceae isolated from bloodstream infections in hospitals in southern Poland. There are no line numbers which makes the revision process very hard. The writing quality of the manuscript is very bad. Here are some suggestions to improve the manuscript:
- In the abstract, provide numbers in the result part.
- TEM, SHV, and CTX-M: add the full name.
- Add more information in the introduction on ESBLs, AMR in KP and its economic importance.
- materials and methods need to be re-written again.
- In materials and methods: add a title for the first part.
- In materials and methods, you need to provide informations on the collected samples. number, and distribute them
- Divide the materials into sub-sections and follow the previous publications.
- Explain tables and figures then refer to them, not the opposite.
- Write the name of the bacteria in italic.
- Write the full name for the first time in the manuscript then use abbreviations.
- The presented data is not enough for publication, you need to conduct more experiments such as more genotypic analysis, test more antibiotics, and make correlations between phenotypic and genotypic analysis.
Author Response
Lublin, 10.06.2022
Concerns the manuscript Antimicrobial resistance of Enterobacteriaceae in bloodstream infections in hospitalized patients in southern Poland
To whom it may concern,
We would like to thank the Reviewer for the time they devoted to the improved manuscript and preparation of comments - all of them were reconsidered.
Below is the full answer to the doubts of the reviewer.
Yours faithfully,
Marta KÅ‚os, Estera Jachowicz, Monika Pomorska-WesoÅ‚owska, Dorota Romaniszyn, Grzegorz Kandzierski, Jadwiga Wójkowska-Mach
„ Moderate English changes required”
Reply: The English language was checked.
”Does the introduction provide sufficient background and include all relevant references?”
- Add more information in the introduction on ESBLs, AMR in KP and its economic importance.
Reply: The introduction was redrafted and corrected. In the Introduction section, we emphasize the problem of nosocomial infections, including bloodstream infections, increasing mortality, and morbidity. Then we discuss the issue of increasing multidrug resistance of Enterobacterales caused by various factors: inappropriate prophylaxis, lack of effective antimicrobials, failure of empiric therapy.
“Are all the cited references relevant to the research?”
Reply: The references cited were checked.
“Are the methods adequately described?”
- materials and methods need to be re-written again.
- In materials and methods: add a title for the first part.
- In materials and methods, you need to provide informations on the collected samples. number, and distribute them.
- Divide the materials into sub-sections and follow the previous publications.
Reply: Thank you for your note. The materials and methods were corrected. Errors were fixed. Due to the short description of the Materials and Methods section, we did not want to add sub-titles or sub-sections. The Materials and Methods section consists of only two paragraphs, where in the first one we described the basis on which we defined bloodstream infections, years of sampling and their origin. Of course, each of the samplings was given its own individual number. In the second paragraph, we describe the methodology during material collection, microbial identification, and drug resistance analysis. The only sub-title is statistical analyzes described in detail in the Materials and Methods section. Ethic code was added.
“Are the results clearly presented?”
Reply: the Results section was improved. The results, after being improved, describe only Enterobacterales antimicrobial resistance with an emphasis on ESBL strains. Only Table 1 presents an overview of the etiology of bloodstream infections among the group of patients studied. Only enzymes from the SHV, CTX-M and TEM groups are mentioned among the ESBL mechanisms. Tables 2 and 3, with the prevalence of ESBL strains, are intended to show their percentage share in individual units.
“Are the conclusions supported by the results?”
Reply: The conclusions were improved.
Comments and suggestions for authors:
- In the abstract, provide numbers in the result part.
Reply: In the abstract, the numbers were added in the results part.
- TEM, SHV, and CTX-M: add the full name.
Reply: Thank you for your note. We did not want to add the full name of the enzymes so as not to blind the message and to make the manuscript as readable as possible; therefore, we wrote that these are enzymes from the β-lactamase group.
- Explain tables and figures, then refer to them, not the opposite.
Reply: Thank you for your note; it was corrected.
- Write the name of the bacteria in italic.
Reply: Thank you for your note; it was corrected.
- Write the full name for the first time in the manuscript then use abbreviations.
Reply: Thank you for your note; it was corrected. In each section, we start with the full name and then write the abbreviations.
- The presented data is not enough for publication, you need to conduct more experiments such as more genotypic analysis, test more antibiotics, and make correlations between phenotypic and genotypic analysis.
Reply: The presented data show a high resistance of Enterobacterales and their increasing prevalence in bloodstream infections among the studied group of patients. They are also an introduction to further research on drug resistance, genotypic analysis, and their correlation.

Reviewer 4 Report
Overall, this is a well-written, precise, and clear manuscript. The introduction is relevant and theoretical in nature. The prior research findings are described in sufficient detail for readers to understand the current study rationale. The writers provide a systematic contribution to the scientific literature in this field of research. Overall, this is a good manuscript. It requires minor adjustments. Specific remarks are provided below.
A: Introduction:
Bloodstream infections (BSIs) need to be written.
'healthcare-associated infection (HCAI)' highlighted column needs to be rewritten
Healthcare-associated infections need to be written.
B: Materials and Methods:
In 5th line s: 1) semicolon needs to be checked again
In the second paragraph, EUCAST 2022 guidelines are used but the sample collection is 2015-2018 recheck it again.
if Cochran's condition was met, but if the condition was not met, (this sentence needs clearance).
C: Results:
In the 3rd line table, 1 needs to write without italic form.
D: Discussion:
Instead, BSI (BSIs) needs to be written.
Kaur et al. [ ref. 19] has performed an original double disc synergy test, phenotypic disc confirmatory test (PDCT) and modified double disc synergy test (MDDST) for comparisons of the results for ESBL production. So, what tests other than the double-disc synergy test you have performed for your confirmation and comparison other than in silico. Needs to explain?
Author Response
Lublin, 10.06.2022
Concerns the manuscript Antimicrobial resistance of Enterobacteriaceae in bloodstream infections in hospitalized patients in southern Poland
To whom it may concern,
We would like to thank the Reviewer for the time they devoted to the improved manuscript and preparation of comments - all of them were reconsidered.
Below is the full answer to the doubts of the reviewer.
Yours faithfully,
Marta KÅ‚os, Estera Jachowicz, Monika Pomorska-WesoÅ‚owska, Dorota Romaniszyn, Grzegorz Kandzierski, Jadwiga Wójkowska-Mach
”Does the introduction provide sufficient background and include all relevant references?”
A: Introduction:
Bloodstream infections (BSIs) need to be written.
'healthcare-associated infection (HCAI)' highlighted column needs to be rewritten
Healthcare-associated infections need to be written.
Reply: The introduction was redrafted and corrected. In the first paragraph of the Introduction Section, we describe hospital-acquired infections and healthcare-associated infections, while bloodstream infections are described in the second paragraph of the Introduction Section and in the Materials and Methods Section. In the Introduction section, we emphasize the problem of nosocomial infections, including bloodstream infections, increasing mortality, and morbidity. Then we discuss the issue of increasing multidrug resistance of Enterobacterales caused by various factors: inappropriate prophylaxis, lack of effective antimicrobials, failure of empiric therapy.
“Are the methods adequately described?”
B: Materials and Methods:
In 5th line s: 1) semicolon needs to be checked again
In the second paragraph, EUCAST 2022 guidelines are used but the sample collection is 2015-2018 recheck it again.
if Cochran's condition was met, but if the condition was not met, (this sentence needs clearance).
Reply: Thank you for your note. The materials and methods were corrected. The errors were fixed. We used the colon first because we are listing the inclusion criteria of secondary BSIs, and each listed criterion is followed by a semicolon. During the study period, 2015-2018, we used the 2017 EUCAST guidelines. The article was written in 2022, so we compared these data and used the 2022 EUCAST guidelines. If the condition was not met, Fisher's exact test was used. Ethic code was added.
“Are the results clearly presented?”
C: Results:
In the 3rd line table, 1 needs to write without italic form.
Reply: The Results section was improved.
Comments and suggestions for authors:
D: Discussion:
Instead, BSI (BSIs) needs to be written.
Reply: the discussion section was improved.
Kaur et al. [ ref. 19] has performed an original double disc synergy test, phenotypic disc confirmatory test (PDCT) and modified double disc synergy test (MDDST) for comparisons of the results for ESBL production. So, what tests other than the double-disc synergy test you have performed for your confirmation and comparison other than in silico. Needs to explain?
Reply: To detect and confirm the production of ESBL, we carried out the double disc synergy test and detecting the bla genes using multiplex PCR.

Round 2
Reviewer 3 Report
The conducted experiments are not enough. You need to do more experiments. You need to do phenotypic analysis for the resistant genes and/or sequencing
Author Response
Lublin, 15.06.2022
Concerns the manuscript Antimicrobial resistance of Enterobacteriaceae in bloodstream infections in hospitalized patients in southern Poland
To whom it may concern,
We would like to thank the Reviewer for the time they devoted to the improved manuscript and preparation of comments - all of them were reconsidered.
Below is the full answer to the doubts of the reviewer.
Yours faithfully,
Marta KÅ‚os, Estera Jachowicz, Monika Pomorska-WesoÅ‚owska, Dorota Romaniszyn, Grzegorz Kandzierski, Jadwiga Wójkowska-Mach
“Moderate English changes required”
Reply: The English language and style were checked and corrected.
”Does the introduction provide sufficient background and include all relevant references?”
Reply: The introduction was rewritten and corrected. In the first paragraph of the Introduction Section, we describe hospital-acquired infections and healthcare-associated infections, while bloodstream infections are described in the second paragraph of the Introduction Section and in the Materials and Methods Section. In the Introduction section, we emphasize the problem of nosocomial infections, including bloodstream infections, increasing mortality, and morbidity. Then we discuss the issue of increasing multidrug resistance of Enterobacterales caused by various factors: inappropriate prophylaxis, lack of effective antimicrobials, failure of empiric therapy.
“Is the research design appropriate?”
Reply: Thank you for your note. The research design was designed according to the objective set at the beginning of the study period. The aim was to analyse antimicrobial resistance among Enterobacterales isolated from bloodstream infections in hospitals with a special emphasis on ESBL. The next steps and guidelines were followed. First, the research material was collected from hospitalized patients with an episode of bloodstream infections. The isolates were identified. Next, the presence of the bla genes was determined and antimicrobial susceptibility testing was performed. Each step is described in detail in the Materials and Methods section, according to the guidelines or the manufacturer's instructions. All guidelines were followed and the highest standards were observed.
“Are the methods adequately described?”
The conducted experiments are not enough. You need to do more experiments. You need to do phenotypic analysis for the resistant genes and/or sequencing
Reply: Thank you for your note. The Materials and Methods Section was corrected. The errors were fixed. For the purposes of our work, each strain has been described phenotypically. We conducted a detailed antimicrobial resistance analysis using the disc diffusion method and the double disc synergy test to detect ESBL-producing strains. Since we focus on ESBL-producing strains, we also performed multiplex PCR analysis to detect and confirm the presence of the bla genes.
“Are the results clearly presented?”
Reply: The Results section was improved. In the text, the results for Gram-positive bacteria have been deleted - only the results for Gram-negative bacteria are presented.
